# Meckel’s Diverticulum in Children: A Monocentric Experience and Mini-Review of Literature

**DOI:** 10.3390/children9010035

**Published:** 2022-01-01

**Authors:** Matthias Nissen, Volker Sander, Phillip Rogge, Mohamad Alrefai, Ralf-Bodo Tröbs

**Affiliations:** 1Department of Pediatric Surgery, Marien Hospital Witten, St. Elisabeth Gruppe, Ruhr-University of Bochum, Marienplatz 2, D-58452 Witten, Germany; volker.sander@elisabethgruppe.de (V.S.); phillip.rogge@elisabethgruppe.de (P.R.); mohamad.alrefai@elisabethgruppe.de (M.A.); 2Department of Pediatric Surgery, St. Johannes Hospital, Helios Group, An der Abtei 7 bis 11, D-47166 Duisburg, Germany; troebs@icloud.com

**Keywords:** pediatric vitelline duct anomalies, Meckel’s diverticulum, gastric heterotopia, intussusception, enteric hemorrhage

## Abstract

Vitelline duct anomalies (VDA, including Meckel’s diverticulum (MD)) result from failed embryologic obliteration. This study aimed for characteristics in symptomatic versus asymptomatic VDA, analyzing clinico-laboratory data from 73 children, aged 1 day to 17 years, treated at a tertiary Pediatric Surgery Institution from 2002–2017. A male preponderance was obtained (ratio 3.6:1). MD accounted for 85% of VDA. Incidence of symptomatic VDA decreased with older age. Leading symptoms were intestinal obstruction and hemorrhage. Mucosal heterotopia (present in 39% of symptomatic MD) was associated with anemia and lowered CRP-levels. On ROC-analysis, hemoglobin < 8.6 g/dL, CRP < 0.6 mg/dL and MD distance to ileocecal valve >40 cm were predictors of ectopic tissue in symptomatic MD. Our data confirmed known characteristics as male preponderance, declined incidence of symptomatic cases with age and predominance of gastric ectopia in symptomatic MD. Moreover, anemia and prolonged distance of MD to ileocecal valve were predictors of ectopic mucosa in symptomatic MD.

## 1. Introduction

Failure in obliteration of the vitelline (or omphalomesenteric; OMD) duct, that connects the yolk sac to the embryonic midgut by the seventh to eighth gestational week, results in various forms of vitelline duct anomalies (VDA) [1,2]. Depending on the extent and site of disintegration and absorption, several anomalies may result including patent OMD, fibrous cords from umbilicus to ileum, omphalomesenteric (umbilical) duct sinus or cyst in case of its distal persistence. However, its proximal part persistence constitutes the most commonly observed representative of VDA, known as Meckel’s diverticulum (MD) [2,3]. The prevalence of MD varies from 0.3% to 2.9% with a male to female ratio of 1.5:1 to 4:1 [2,4]. In the majority of cases, MD is clinically silent, only incidentally identified by radiographic imaging, endoscopy or surgery for another reason [2]. The estimated lifetime risk of a symptomatic MD is about 4.2%, decreasing towards zero with increasing age [5,6]. MD usually presents a normal small bowel wall architecture, lined by ileal mucosa. However, ectopic stomach mucosal lining may be found in 4.6% to 71% of symptomatic MD, followed by ectopic nodules of pancreatic tissue in up to 12% and, in rare cases, by colonic and duodenal mucosa [4]. The presence of ectopia is associated with symptomatic MD in general and intestinal bleeding in particular [7]. MD most commonly comprises intestinal obstruction, hemorrhage or diverticulitis [8]. Obstruction refers to cases of mechanical compromise by intussusception with MD as leading point, small intestine volvulus around the diverticular axis or persisting cords, while intestinal hemorrhage occurs secondary to ulcers in the adjacent intestinal tissue, caused by ectopic gastric acid secretion, often presenting as painless bleeding per rectum. Lastly, inflammation refers to inflamed diverticula with or without perforation, eventually resulting in peritonitis [4]. Given that VDA in general, and MD in particular, may present a multifaceted clinical spectrum, the purpose of this study was to identify predictive pre- or intraoperative elements. In specific, we were interested in clinico-laboratory and morphological parameters of discriminatory value regarding symptomatic versus asymptomatic VDA and ectopic versus non-ectopic mucosal states of MD.

## 2. Materials and Methods

In a retrospective monocentric review, we identified 73 consecutive cases with VDA or MD (International Classification of Diseases, 10th Revision [ICD-10], code: Q43.0), undergoing surgical verification (*n* = 73) and removal (*n* = 70) between February 2002 and March 2017 at our Department of Pediatric Surgery. If not found incidentally, VDA was assumed clinically and/or through medical history and was confirmed by surgery and/or by histopathology in each case. The estimated parameters included the patients’ biometrical data, morphometric and histological workup of VDA, surgery time, length of stay, symptoms and laboratory data on admission. Data acquisition and basic analysis were performed with Microsoft Excel© (Microsoft Corporation, Redmond, WA, USA). Statistical analysis was performed with OriginPro2021© software (OriginLab, Northampton, MA, USA). Kolmogorov–Smirnov normality test at 0.05-significancy level confirmed normal distribution of numeric variables. For data description, we applied the arithmetic mean ± standard deviation (SD) or median (1st and 3rd quartiles; (Q1–Q3)) in case of non-normally distributed data. Comparative analysis of non-continuous values was evaluated by two sample student’s *t*-testing (±Welch correction) when data were normally distributed (±different variances), otherwise by non-parametric Mann–Whitney testing with *p* ≤ 0.05 as significant. Parametric data of >2 groups were evaluated using one-way analysis of variance (ANOVA) for repeated measurements with a Tukey post hoc test. In addition, homogeneity of variances was evaluated by Levene’s test. Comparisons of non-normally distributed data of >2 groups were performed by Kruskal–Wallis ANOVA testing. Categorical variables were presented as frequencies and percentages and compared using Fisher’s exact test. Furthermore, areas under the curve (AUC) of receiver operating characteristic (ROC) curves were analyzed and sensitivities at 95% specificity were given. ROC curve analysis was performed to determine optimal cut-off for variables of predictive value for presence of ectopic tissue applying the Youden index [9]. Sensitivity, specificity, positive predictive value (PPV), negative predictive value (NPV) and odds ratio (OR) were calculated for the chosen cut-off values. Statistic differences are stated as * (*p* ≤ 0.05), ** (*p* ≤ 0.01) or *** (*p* ≤ 0.001).

## 3. Results

Diagnosis of VDA was surgically verified in 73 patients. There were 57 males and 16 females (male-to-female ratio 3.6:1; Table 1). The age ranged from 1 day to 17.1 (mean 4.7 ± 5.3) years.

### 3.1. Age

Age-stratified characteristics of VDA are given in Table 1. In 62%, VDA was diagnosed in children younger than 5 years. MD comprised 85% of VDA. OMD was most frequently diagnosed in neonatal age class (50%). Ectopic tissue was found in ≥30% within each pediatric age group older than 1 year and was predominantly of gastric origin (overall gastric-to-pancreatic ratio 13:1). Length and base width of MD were even among age classes. However, ileocecal valve distance of MD in tendency was increasing with older ages. Symptomatic VDA peaked (78%) at age class 28 days–1 year, gradually decreasing with increased age, reaching 47% in age class >10 years. Etiology of symptomatic cases was led by intestinal obstruction (41%), followed by bleeding (27%) and diverticulitis/others (both 16%). Postsurgical length of stay (LOS) and surgery time did not differ between groups.

### 3.2. Symptomatology

Dichotomization into asymptomatic (*n* = 29) and symptomatic (*n* = 44) cases was performed with further subdivision of the latter into intestinal obstruction (*n* = 18), bleeding (*n* = 12), diverticulitis (*n* = 7) and others (*n* = 7) (Table 2). Regarding asymptomatic cases, main reason for surgery was appendicitis (*n* = 15), followed by meconium ileus, omphalocele and ostomy closure (each *n* = 3), and lastly, one case each for alimentary tract duplication, heminephrectomy, cryptorchism, intestinal atresia, infantile hypertrophic pyloric stenosis, focal intestinal perforation and ventriculoperitoneal shunt infection in a child with hydrocephalus due to peritonitis secondary to appendicitis. Intestinal obstruction group comprised ten cases of intussusception with MD as a pathological lead point, followed by strangulation (*n* = 4); volvulus (*n* = 2), internal hernia (closed loop obstruction; *n* = 1) and perforation secondary to obstruction without inflammation (*n* = 1). Comparison of asymptomatic versus symptomatic cases revealed a relatively decreased hemoglobin level *(p* = 0.037). However, at ROC analysis, hemoglobin failed being of discriminatory value for symptomatic VDA (AUC 0.669 ± 0.10; 95% CI 0.48–0.86; *p* = 0.07). Regarding subgroup analysis, hemoglobin was lower in gastrointestinal hemorrhage compared to the obstruction group *(p* = 0.001). Moreover, although within normal value range, a relatively elevated CRP *(p* = 0.009) was obtained for obstruction and diverticulitis compared to bleeding/others group. While being present in 70% of available data on intestinal hemorrhage, mucosal heterotopia was present in only 27% of available data on intestinal obstruction.

### 3.3. Tissue Heterotopia

Data on histopathological proven heterotopia (39%) and non-heterotopia (61%) was available for 28 symptomatic patients with MD (Table 3). In ten cases, heterotopic mucosa was of gastric, and in one case of pancreatic origin. Regarding heterotopia, values on age (*p* = 0.39), body weight (*p* = 0.24) and body length (*p* = 0.09) on admission showed a trend towards elevation, whereas hemoglobin level (*p* = 0.01), CRP (*p* = 0.019) and presence of vomiting (*p* = 0.003) on admission were lower in cases of heterotopic mucosa. Moreover, distance of MD to ileocecal valve (*p* = 0.047) and presence of intestinal bleeding (*p* = 0.02) were higher in heterotopic mucosa. To determine the value of parameters regarding discrimination of ectopia from non-ectopia in symptomatic MD, ROC-analysis was applied. Hemoglobin < 8.6 g/dL (AUC 0.837 ± 0.101 (95% CI 0.64–1.03); *p* = 0.014; sensitivity 72%; specificity 93%; positive predictive value (PPV) 83%; negative predictive value (NPV) 87%; odds ratio (OR) 32.5 (95% CI 2.4–443.2)), CRP < 0.6 mg/dL (AUC 0.854 ± 0.095; (0.67–1.04); *p* = 0.017; sensitivity 83%; specificity 58%; PPV 50%; NPV 88%; OR 7 (0.61–79.9)) and distance of MD diverticulum from ileocecal valve >40 cm (AUC 0.889 ± 0.111 (0.67–1.11); *p* = 0.009; sensitivity 86%; specificity 89%; PPV 86%; NPV 89%; OR 48 (2.5–932.9)) were of predictive value for presence of ectopic tissue.

## 4. Discussion

In this study, we confirmed known characteristics for VDA and MD as male preponderance, a decline of symptomatic cases with age and gastric mucosal ectopia that is predominantly found in symptomatic cases, that also constitutes a prerequisite for hemorrhage. Moreover, we identified anemia and prolonged distance of MD from ileocecal valve as predictive factors associated with presence of ectopic tissue.

### 4.1. Gender

In accordance with literature, a male preponderance was observed throughout all age-groups of our cohort [1,2,4,8]. In symptomatic cases, this predominance was even more pronounced, rising from 2.2:1 in asymptomatic cases to 5:1 and above (Table 2). As proposed by Huang et al. in a retrospective study on 100 cases with MD, elevated male acid-production and therefore an elevated risk for peptic ulcer might account for this observed gender-specific distribution [1]. In addition, they hypothesized that the Cdx2-homebox transcription factor might play a pivotal role in determining the intestinal epithelial phenotype during embryogenic development, a mechanism possibly accounting for the prominence of male gender in heterotopic MD. In this context, Martin et al. described a case of absent Cdx2 expression in congenital gastric-type heteroplasia in the mid-jejunum of an eight years old male, which was similar to the gastric heteroplasia in the developing gut endoderm observed in a murine model with loss of function of the homeobox gene Cdx2 [10]. Moreover, as observed in gastric cancer patients, methylation of Cdx2 with consecutive downregulation of gene expression is predominantly found to be increased in males, possibly being ascribed to gender-specific host factors such as estrogens [11].

### 4.2. Age

In line with literature data [12], compared with age at manifestation of intestinal hemorrhage (5.6 ± 5.8 years; Table 2), patients were (marginally) younger in cases of obstruction (4.3 ± 5.3 years) and diverticulitis (4.5 ± 5.9 years). These age-differences are in line with implications derived from spatial variations within density pattern of myenteric plexuses in MD, as observed by Negrea et al. in a small study on eight cases [13]. Compared with ectopic gastric mucosa-lined areas, they found an increased density of myenteric nerve fibers in areas with enteric mucosa. It is suggested that nerve fiber density variations determine local peristalsis. MD-expressing enteric type mucosa with their higher peristaltic activity are more susceptible for intussusception or, in case of intraluminal obstruction, favor worsening of infectious state and perforation. On the other hand, MD with gastric heterotopia expressing lower density of Auerbach’s plexus nerve fibers, might be less effective in its content drainage, favoring prolonged enteric mucosa contact with MD’s acid secrete and thus development of peptic ulceration adjacent to the gastric mucosa zone [13]. Given these different histomorphological patterns in MD, two factors might explain delayed manifestation of hemorrhage compared with diverticulitis and obstruction in particular, and the observed dissolution of symptomatic MD with age in general. First, acid production in ectopic gastric mucosa seems to be increased with age, as described in a study including 47 children by Tseng and Yang [12]. Second, available data on myenteric neuron density suggests an underlying age-dependent progressive decrease [13].

### 4.3. Symptomatology

With a proportion of 60% on symptomatic cases (Table 2), our data are in concordance with findings by others [14,15]. Moreover, with only 11%, a decreased prevalence of heterotopia in asymptomatic patients was evident, contrasting 31% of symptomatic cases (Table 2). Our laboratory findings with a lowered CRP and hemoglobin levels in bleeding group compared to obstruction and diverticulitis group are confirmed by Parvanescu et al. on 37 complicated adult cases with MD [8]. Noteworthy, observed variations in CRP were within normal value limits. Moreover, observed lowered hemoglobin levels in the symptomatic group (*p* = 0.037) failed being of predictive value (AUC 0.669 ± 0.10; *p* = 0.07). The incidence of symptomatic VDA decreased with age. In our series, 84% of symptomatic cases were aged ≤10 years and 68% ≤ 5 years. This is in line with literature data, stating that more than half of symptomatic patients are below ten years of age [4]. Regarding blood transfusion, half of our cases with intestinal hemorrhage required at least one blood transfusion during hospital stay. This is in line with frequencies documented by others, ranging from 67–71% of patients with intestinal hemorrhage secondary to MD [12,16].

### 4.4. Tissue Heterotopia

Symptomatic cases with heterotopic tissue were associated with lowered hemoglobin levels and larger distances between MD and the ileocecal valve (Table 3), both being of fair to good discriminatory accuracy. Our findings of an enlarged distance between MD and the ileocecal valve in cases of heterotopia might either reflect proportional intestinal growth with age, or is an independent factor associated with heterotopia itself. These age dependent variations in distance between diverticulum and the ileocecal valve were also found by Yamaguchi et al., who investigated 113 patients, in which mean distance was gradually increased from patients aged below two years to patients aged above 21 years [17]. The tendency towards an older age on presentation with heterotopia (6.5 ± 6.0 vs. 4.5 ± 5.8 years; *p* = 0.24) was indirectly supported by Burjonrappa and Khaing, who stated that symptomatic children with ectopic tissue were older than those without ectopic tissue (mean 6.3 vs. 3.1 years) [18]. The following findings by Ban-Hani and Shatnawi [19] are in accordance with our results. Symptomatic MD with proven gastric ectopic tissue was present in 32% (31% in our series; Table 2) of their available histology reports, while pancreatic tissue was found in 4% (3% in our series). Moreover, of analyzed non-symptomatic cases, 10% (11% in our series) had ectopic gastric tissue without expression of pancreatic tissue in either study. Given an almost three-times lowered incidence of heterotopia in asymptomatic MD, our sample size might be too low for elucidating reliable factors in support of decision making whether incidentally found MD might be removed or retained during surgery. A palpable thickening of the MD has been previously proposed as an indicator of ectopic tissue [4]. In addition, morphologic characteristics such as increased diameter of MD basis alone [20], or a decrease in the height-to-diameter ratio [21] have been proposed indicators of present heterotopia. However, our morphometric data on MD were not altered within groups. In our study, one case of symptomatic MD (9%) was of pancreatic origin and associated with intussusception. Correspondingly, according to the literature, pancreatic heterotopia is present in up to 12% of symptomatic MD and might also be considered a lead point for intussusception [1,4]. The uniqueness of heterotopic gastric mucosa in some cases of MD is utilized as a diagnostic tool. Ectopic gastric mucosa is a prerequisite for its detection by scintigraphy of the isotope Tc99 sodium pertechnetate (Meckel’s scan) that has a strong affinity for parietal cells of both eutopic and ectopic gastric mucosa [2]. However, Meckel’s scan sensitivity varies from 25% to 92% with a 50% false negative rate in hemoglobin values below 11 g/dL [2]. In our series, three cases with histologically proven symptomatic MD underwent Meckel’s scan (27%), of whom the result was true positive in two cases (confirmed gastric ectopia) and false negative (non-confirmed ectopia) in one case (Table 3). Regarding modern aspects of diagnostics, capsule endoscopy has been recently introduced as a promising tool for the diagnosis of MD [22]. However, to date, this technique has only been reported in smaller case series [23,24] and one larger study [25]. Although performed in 91% of cases with ectopic mucosa, no case of MD was assumed by ultrasound (US). In cases of non-ectopic tissue, US was utilized in 71% of cases with preoperatively suspected MD in one case (8%) and intussusception in eight cases (67%). By preoperative X-ray, associated ileus was confirmed in 36% and 35% of ectopic and non-ectopic tissue cases (Table 3).

### 4.5. Surgery

Initial laparoscopy was utilized in 49% with a conversion rate of 81% into (mini-)laparotomy by vertical extension of the infra-umbilical laparoscopic access (Table 1). However, primary laparotomy was performed in 51% of cases. In 73% and 82% of ectopic and non-ectopic cases, segmental ileal resection was performed (Table 3). Surgical treatment of symptomatic MD is mandatory. Besides the classical open procedure, laparoscopy has been increasingly utilized [21,26] and may be considered the current modality that is most often leading to correct diagnosis of MD [22]. Considering the more distal distribution of gastric heterotopia in long diverticula and a more arbitrary distribution of gastric mucosa in short diverticula, simple (laparoscopic) transverse resection of long MD by linear stapler must be weighed against (extracorporal) wedge-shaped excision or segmental ileal resection, as proposed in cases of short diverticula or hemorrhage with bleeding site that is commonly located at the adjacent ileal wall [1,21]. Presence of heterotopic tissue is still a microscopic determinant, and no morphologic parameter with high accuracy for identification of ectopic tissue exists. Therefore, current proposed management in children tends to be excision of MD by segmental bowel resection rather than a simple diverticulectomy. This is supported by a recent systematic review, estimating a relative risk of 3.64 (95% CI 3.1–4.3) for becoming a symptomatic MD in case of present gastric heterotopia [5]. Currently, preventive resection of a silent MD, especially in the older children, is an issue of controversial debate. A 5.3% risk of postoperative complications following prophylactic resection, as reported by Zani et al., is weighted against the risk of 1.3% in developing symptoms secondary to non-resected MD [5]. Zani et al. further estimated that 750 cases of non-symptomatic MD must be resected in order to save one life. Accordingly, Soltero and Bill stated that, in order to prevent one case of death, removal of 800 incidental MD would be necessary [6]. They therefore suggested that incidental MD removal is not justifiable unless risk factors such as meso-diverticular bands or fibrous bands to the umbilicus are obvious. Park et al. performed a large retrospective study on 1476 patients throughout all ages and identified four factors inherent to an elevated likelihood of a silent MD becoming symptomatic, advocating its prophylactic resection, namely male gender, age < 50 years, diverticular length > 2 cm and presence of ectopic tissue [27]. By finding of a more evenly distributed rather than a decreased risk of developing symptomatic MD with age, Cullen et al. also recommended prophylactic resection, provided that no contraindications like generalized peritonitis or advanced inflammatory states of appendicitis are present [28].

### 4.6. Limitations

One major limitation of this study is its small sample size. Another restriction is our single-institutional, retrospective study design. Since selection criteria were based on diagnostic coding, we might have missed concurrent incidental occurrence of VDA during surgical procedures for other reasons in which VDA might have been present but not coded for. Another source of bias can be found in the historic character of our cohort itself with its high rate of incidentally resected MD that is not necessarily reflecting current knowledge on this disease. In specific, our comparable high rate of incidentally resected MD (40% of study cohort) reflects a period where therapy and strategies in surgical treatment of MD were changing. According to the actual literature, not all of these cases would have been treated by surgery nowadays.

## 5. Conclusions

In this study, we identified known characteristics for VDA such as male preponderance, predominant gastric ectopia in symptomatic MD constituting a prerequisite for hemorrhage and an age-dependent decline of symptomatic cases. Moreover, anemia and prolonged distance of MD from ileocecal valve were identified predictive factors for the presence of ectopic tissue in symptomatic MD. Diagnosis of VDA and MD remains challenging and, given the lack of sustainable diagnostic radiologic tools, surgical exploration and (preferably laparoscopic) excision of symptomatic MD is widely recommended in children while treatment of incidentally encountered silent MD, especially in older children, is basically conservative. In conclusion, manifestation of VDA in children is highly variable and dependent on patients’ age and presence of associated heterotopic tissue.

## Figures and Tables

**Table 1 children-09-00035-t001:** Age-stratified characteristics of vitelline duct anomalies.

Age Groups.	All	≤28d	>28d–1a	>1a–5a	>5a–10a	>10a	*p*
Demographic							
Patients	73 (100)	12 (16)	18 (25)	15 (21)	13 (18)	15 (21)	
Age at surgery (d)	1709 ± 1949	8.9 ± 8.6	157 ± 83	914 ± 414	2533 ± 492	5011 ± 758	
Gender (m/f)	57/16	10/2	14/4	11/4	11/2	11/4	0.93
Male-to-female ratio	3.6:1	5:1	3.5:1	2.8:1	5.5:1	2.8:1	
Entity							
OMD	11 (15)	6 (50)	2 (11)	2 (13)	1 (8)	0	
MD	62 (85)	6 (50)	16 (89)	13 (87)	12 (92)	15 (100)	
MD base width (cm)	1.6 ± 0.8	n.a.	1.8 ± 1.2	1.3 ± 0.7	1.7 ± 0.6	1.7 ± 0.7	0.51–0.99
MD length (cm)	2.8 ± 1.3	2.0 ± 1.5	2.4 ± 1.7	2.7 ± 1.3	3.5 ± 1.4	3.0 ± 0.7	0.59–0.99
Ileocecal valve distance (cm)	36 ± 23	19 ± 7	32 ± 18	39 ± 26	46 ± 27	36 ± 16	0.23–0.99
VDA retained in situ	3 (4)	1 (8)	0	1 (7)	0	1 (7)	
Ectopic tissue							
Unknown	17	4	7	4	1	1	
Available histologic data	53	7	11	10	12	13	
Normal mucosa	39 (74)	7 (100)	8 (73)	7 (70)	8 (67)	9 (69)	
Gastric mucosa	13 (24)	0	2 (18)	3 (30)	4 (33)	4 (31)	
Pancreatic mucosa	1 (2)	0	1 (9)	0	0	0	
Clinical							
Incidental	29 (40)	7 (58)	4 (22)	4 (27)	6 (46)	8 (53)	
Symptomatic	44 (60)	5 (42)	14 (78)	11 (73)	7 (53)	7 (47)	
Intestinal bleeding	12 (27)	0	1 (7)	7 (64)	1 (14)	3 (43)	
Intestinal obstruction	18 (41)	1 (20)	8 (57)	2 (18)	4 (57)	3 (43)	
Diverticulitis	7 (16)	0	3 (21)	1 (9)	2 (29)	1 (14)	
Others	7 (16)	4 (80) ^†^	2 (14) ^‡^	1 (9) ^§^	0	0	
Procedural							
Postsurgical length of stay (d)	10 ± 7	12 ± 11	10 ± 6	9 ± 5	9 ± 5	11 ± 9	0.78–0.99
Surgery time (min)	144 ± 68	124 ± 53	156 ± 58	138 ± 65	147 ± 76	151 ± 86	0.72–0.99
Initial laparoscopy	36 (49)	0	6 (33)	8 (53)	8 (62)	14 (93)	
Conversion to laparotomy	29 (81)	0	6 (100)	8 (100)	5 (63)	10 (71)	
Initial laparotomy	37 (51)	12 (100)	12 (67)	7 (47)	5 (39)	1 (7)	

OMD Omphalomesenteric duct; MD Meckel’s diverticulum, VDA Vitelline duct anomaly, n.a. not available, Umbilical granuloma: ^†^ (*n* = 4), ^‡^ (*n* = 2), ^§^ Fibrous cord remnant (*n* = 1). Categorical variables were presented as frequencies and percentages (in brackets) and compared by Fisher’s exact test; *p* ≤ 0.05 was defined significant.

**Table 2 children-09-00035-t002:** Characteristics of asymptomatic and symptomatic vitelline duct anomalies and Meckel’s diverticula.

	Non-Symptomatic		Symptomatic												
		*n*	All	*n*	*p*		Obstruction	*n*	Bleeding	*n*	Diverticulitis	*n*	Others	*n*	*p*
Demographic															
Patients *n* (%)	29 (40)		44 (60)				18 (41)		12 (27)		7 (16)		7 (16)		
Age (y)	5.6 ± 5.5	29	4.1 ± 5.2	44	0.63		4.3 ± 5.3	18	5.6 ± 5.8	12	4.5 ± 5.9	7	0.5 ± 1.0	7	0.18–0.99
Gender (m/f)	20/9		37/7		0.15		15/3		10/2		6/1		6/1		0.93
Male-to-female ratio	2.2:1		5.3:1				5:1		5:1		6:1		6:1		
Laboratory															
Hb (g/dL)	13.3 ± 2.0	15	11.5 ± 3.6	29	0.037		13.5 ± 3.2	12	7.9 ± 3.1	8	11.8 ± 2.0	5	12.3 ± 2.1	4	0.001 ^†^; all others: 0.09–0.99
WBC (10^9^/L)	13.9 ± 8.3	16	13.6 ± 7.8	28	0.93		12.4 ± 4.9	13	11.1 ± 2.8	6	21.1 ± 15.2	5	12.1 ± 3.9	4	0.14–0.99
CRP (mg/dL)	1.6 (0.1–7.3)	18	0.5 (0.0–0.8)	25	0.29		0.5 (0.1–4.3)	11	0 (0–0.2)	6	0.8 (0.4–4.5)	5	0.0 (0–0.5)	3	0.009
Entity															
MD	27 (93)		35 (80)				17 (84)		12 (100)		6 (86)		0		
OMD	2 (7)		9 (21)				1 (6)		0		1(14)		7 (100)		
VDA not excized	3 (10)		n/a				n/a		n/a		n/a		n/a		
Ectopic tissue															
Unknown	8 (28)		9 (21)				3 (17)		2 (17)		2 (29)		2 (29)		
Available data	18		35				15		10		5		5		
None	16 (89)		23 (66)				11 (73)		3 (30)		5 (100)		4 (80)		
Gastric	2 (11)		11 (31)				3 (20)		7 (70)		0		1 (20)		
Pancreatic	0		1 (3)				1 (7)		0		0		0		

WBC White blood cell count, CRP C-reactive protein, MD Meckel’s diverticulum, OMD Omphalomesenteric duct; VDA Vitelline duct anomaly, ^†^ Obstruction versus bleeding group. n/a not applicable; Categorical variables were presented as frequencies and percentages (in brackets) and compared by Fisher’s exact test; *p* ≤ 0.05 was defined significant.

**Table 3 children-09-00035-t003:** Ectopic versus non-ectopic tissue finding in symptomatic Meckel’s diverticula.

	Ectopic Tissue		Non-Ectopic Tissue		
		*n*		*n*	*p*
Demographic					
Patients *n* (%)	11 (39)		17 (61)		
Age at surgery (y)	6.5 ± 6.0	11	4.5 ± 5.8	17	0.39
Weight (kg)	27.0 ± 25.1	8	16.2 ± 18.0	16	0.24
Length (cm)	120 ± 48	6	83 ± 31	9	0.09
Gender (m/f)	8/3		14/3		0.65
Male-to-female ratio	2.7:1		4.7:1		
Laboratory					
Hemoglobin (g/dL)	8.0 ± 2.6	7	12.6 ± 3.8	14	0.010
WBC (10^9^/L)	12.3 ± 3.9	6	13.5 ± 10.0	14	0.32
CRP (mg/dL)	0.1 (0–0.6)	6	0.7 (0.2–3.6)	12	0.019
Platelets (10^9^/L)	353 ± 110	5	402 ± 157	14	0.54
MD morphometry					
MD base width (cm)	1.3 ± 0.7	5	1.7 ± 0.9	12	0.27
MD length (cm)	3.5 ± 1.6	6	2.8 ± 1.3	7	0.37
MD length-to-width ratio	2.6 ± 1.3	5	1.4 ± 0.6	6	0.07
Ileocecal valve distance (cm)	57 ± 21	7	29 ± 28	24	0.047
Etiology					
Intestinal bleeding	7 (64)		3 (18)		0.020
Intestinal obstruction	4 (36)		10 (59)		0.44
Diverticulitis	0		5 (29)		0.13
Clinical presentation					
Vomiting	3 (27)		15 (88)		0.003
Bleeding	7 (64)		3 (18)		0.020
Abdominal pain	7 (64)		12 (71)		0.29
Fever	1 (9)		3 (18)		1
Septic	0		1 (6)		1
Ileus	4 (36)		10 (59)		0.44
Procedural					
Postsurgical length of stay (d)	7 ± 3	11	11 ± 6	17	0.051
Surgery time (min)	126 ± 55	11	144 ± 59	17	0.431
Ileal segmentectomy	8 (73)		14 (82)		0.65
Wedge resection	1 (9)		1(6)		>0.99
Linear stapling	2 (18)		2 (12)		>0.99
Performed US	10 (91) ^†^		12 (71) ^‡^		0.36
Performed Meckel’s scan	3 (27) ^§^		0		0.051
Ileus on X-ray	4 (36)		6 (35)		>0.99

WBC White blood cell count, CRP C-reactive protein, MD Meckel’s diverticulum, VDA vitelline duct anomaly, US Ultrasound, ^†^ No case of VDA assumed, ^‡^ MD assumed (*n* = 1) and intussusception confirmed (*n* = 8), ^§^ Meckel’s scan true (*n* = 2) and false positive (*n* = 1). Categorical variables were presented as frequencies and percentages (in brackets) and compared by Fisher’s exact test; *p* ≤ 0.05 was defined significant.

## Data Availability

The raw data supporting the conclusions of this article will be made available by the authors without undue reservation.

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
