# Peer review of "Meckel’s Diverticulum in Children: A Monocentric Experience and Mini-Review of Literature"

_children, 2022, doi:10.3390/children9010035_

Round 1

Reviewer 1 Report

This is a meticulous review of the characteristics of vitellin duct anomalies (VDA) that were excised from patients over a 15-year period.

The vitellin duct connects the developing intestinal tract with the embryologic yolk sac at the umbilical ring. Normally it disappears as the fetus develops; however, it may persist in whole or in part, and this constitutes the spectrum of vitellin duct anomalies:

  • There may be a patent conduit from the ileum to the umbilicus.
  • The distal part may persist as a mucosal remnant at the umbilicus.
  • The conduit may undergo fibrosis, leaving only a band from ileum to the umbilicus.
  • There may be no attachment to the umbilicus, if only the proximal portion remains; this is termed a Meckel’s Diverticulum (MD).
  • A band may cause intestinal obstruction from internal herniation or volvulus.
  • Meckel’s diverticula may serve as the lead point for intussusception.
  • MD may contain a patch of heterotopic gastric (or pancreatic) mucosa causing ulceration and bleeding, if acid is produced.

Occasionally a MD is found incidentally; and controversy exists, as to whether or not it should be excised.

VDA are fascinating in their variety and the spectrum of their clinical presentations.

The purpose of the paper is to analyze this heterogeneity and find commonality amongst the diversity. The authors painstakingly recorded the length and the width of the diverticula , the distance from the cecum, whether or not there is heterotopic mucosa, what symptoms (if any) were produced.

The authors tabulate the age and sex of patients and the relevant laboratory data; and they thoroughly reviewed the relevant literature.

Many times, their findings are consistent with generally acknowledged attributes of VDA; occasionally they are at variance.

The authors offer some novel conjectures (histologic and genetic) regarding sex distribution and the corresponding clinical presentation: less nerve fibers, causing delay in regional peristalsis allowing acid secretion to concentrate and become ulcerogenic; or more nerve fibers, more vigorous peristalsis + MD leading to intussusception.

The authors are not able to resolve the controversy, regarding what to do with a MD encountered incidentally: leave it alone or excise it; however, they do provide some guidance: since MD are typically symptomatic during childhood, and less so in adults, it is reasonable to remove incidentally encountered MD in children. It may be best to leave them alone in adults.

The paper is thorough and well written; much work went into its preparation; and I believe it should be published. My only caveat is that most of the paper’s conclusions “are in accord with published findings.” 

Author Response

P

Reviewer 2 Report

The authors present a retrospective study on 73 cases with vitelline duct anomalies in children admitted and treated in their department, between 2002 and 2017.  The subject is not new, and the results of the study are generally confirming the previously published articles. However, the paper is well written and clearly structured. The data are rigorously analyzed and their statistical significance is presented.

some minor issues:

  • minor English spelling and styles corrections
  • Conclusions may be shortened in order to reflect the results of the research. For instance, the last phrase should be removed

Author Response

Please see the attached word file…
